# Inverse Versus Normal Behavior of Interactions, Elucidated Based on the Dynamic Nature with QTAIM Dual-Functional Analysis

**DOI:** 10.3390/ijms24032798

**Published:** 2023-02-01

**Authors:** Waro Nakanishi, Satoko Hayashi, Ryosuke Imanaka, Taro Nishide, Eiichiro Tanaka, Hikaru Matsuoka

**Affiliations:** Faculty of Systems Engineering, Wakayama University, 930 Sakaedani, Wakayama 640-8510, Japan

**Keywords:** ab initio calculations, quantum theory of atoms-in-molecules (QTAIM), nonbonded interactions, inverse behavior of interactions, normal behavior of interactions

## Abstract

In QTAIM dual-functional analysis, *H*_b_(***r***_c_) is plotted versus *H*_b_(***r***_c_) − *V*_b_(***r***_c_)/2 for the interactions, where *H*_b_(***r***_c_) and *V*_b_(***r***_c_) are the total electron energy densities and potential energy densities, respectively, at the bond critical points (BCPs) on the interactions in question. The plots are analyzed by the polar (*R*, *θ*) coordinate representation for the data from the fully optimized structures, while those from the perturbed structures around the fully optimized structures are analyzed by (*θ*_p_, *κ*_p_). *θ*_p_ corresponds to the tangent line of the plot, and *κ*_p_ is the curvature; *θ* and *θ*_p_ are measured from the *y*-axis and *y*-direction, respectively. The normal and inverse behavior of interactions is proposed for the cases of *θ*_p_ > *θ* and *θ*_p_ < *θ*, respectively. The origin and the mechanism for the behavior are elucidated. Interactions with *θ*_p_ < *θ* are typically found, although seldom for [F–I-∗-F]^−^, [MeS-∗-TeMe]^2+^, [HS-∗-TeH]^2+^ and CF_3_SO_2_N-∗-IMe, where the asterisks emphasize the existence of BCPs in the interactions and where [Cl–Cl-∗-Cl]^−^ and CF_3_SO_2_N-∗-BrMe were employed as the reference of *θ*_p_ > *θ*. The inverse behavior of the interactions is demonstrated to arise when *H*_b_(***r***_c_) − *V*_b_(***r***_c_)/2 and when the corresponding *G*_b_(***r***_c_), the kinetic energy densities at BCPs, does not show normal behavior.

## 1. Introduction

The quantum theory of atoms-in-molecules dual-functional analysis (QTAIM-DFA) has been proposed to analyze chemical bonds and interactions more effectively [1,2,3] after the QTAIM approach, introduced by Bader [4,5]. The values of QTAIM functions at the bond critical points (BCPs, ∗) on the bond paths (BPs) are often employed for analyses [6]. A chemical bond or an interaction between atoms A and B is denoted by A–B, which corresponds to the BP in the QTAIM. BCP is an important concept in the QTAIM approach, in which charge density *ρ*(***r***) reaches a minimum along the interatomic (bond) path and a maximum on the interatomic surface separating the atomic basins. The *ρ*(***r***) at the BCP is described by *ρ*_b_(***r***_c_), as well as other QTAIM functions, such as total electron energy densities *H*_b_(***r***_c_), potential energy densities *V*_b_(***r***_c_) and kinetic energy densities *G*_b_(***r***_c_) at the BCPs. We use A-∗-B for BP, where the asterisk emphasizes the existence of a BCP in A–B. Equations (1)–(3) show the relationships among the functions (*cf.:* virial theorem for Equation (2)) [4,5].
*H*_b_(***r***_c_) = *G*_b_(***r***_c_) + *V*_b_(***r***_c_)(1)
(*ћ*^2^/8*m*)∇^2^*ρ*_b_(***r***_c_) = *H*_b_(***r***_c_) − *V*_b_(***r***_c_)/2(2)
= *G*_b_(***r***_c_) + *V*_b_(***r***_c_)/2(3)

Interactions are usually classified by the signs of ∇^2^*ρ*_b_(***r***_c_) and *H*_b_(***r***_c_), where the signs of ∇^2^*ρ*_b_(***r***_c_) can be replaced by those of *H*_b_(***r***_c_) − *V*_b_(***r***_c_)/2 because (*ћ*^2^/8*m*)∇^2^*ρ*_b_(***r***_c_) = *H*_b_(***r***_c_) − *V*_b_(***r***_c_)/2 (Equation (2)) [4,5]. They are called shared-shell (SS) interactions when *H*_b_(***r***_c_) − *V*_b_(***r***_c_)/2 < 0 and *H*_b_(***r***_c_) < 0 and closed-shell (CS) interactions for *H*_b_(***r***_c_) − *V*_b_(***r***_c_)/2 > 0. The CS interactions are called pure CS (*p*-CS) for *H*_b_(***r***_c_) − *V*_b_(***r***_c_)/2 > 0 and *H*_b_(***r***_c_) > 0. We proposed to call the interactions with *H*_b_(***r***_c_) − *V*_b_(***r***_c_)/2 > 0 and *H*_b_(***r***_c_) < 0 regular CS (*r*-CS) interactions, which clearly distinguishes them from the *p*-CS interactions.

In QTAIM-DFA, *H*_b_(***r***_c_) is plotted versus *H*_b_(***r***_c_) − *V*_b_(***r***_c_)/2 at BCPs of interactions. Data from the perturbed structures around the fully optimized structures are employed, in addition to the data from the fully optimized structures, in our treatment. Standard interactions have been selected for the CS interactions in vdW adducts (**1**–**4**); typical HBs (*t*-HBs: **5**–**9**) with no covalency (*t*-HB_nc_) and *t*-HBs with covalency (*t*-HB_wc_); molecular complexes formed through charge transfer (CT-MCs: **10**–**15**); TBP adducts formed through CT (CT-TBPs: **16**–**23**) containing trihalide ions (X_3_^−^: **16**–**19**); and the SS interactions of covalent species (Cov: **24**–**36**), both weak (Cov-w: **24**–**29**) and strong (Cov-s: **30**–**36**) (Figure 1). Data for **1**–**36** are collected in Appendix A. Figure 1a shows the plot for **1**–**29,** and Figure 1b shows the plot for **30**–**36**, except for **34**. The plots are well separated, showing that they can be analyzed effectively, and the plots, as a whole, show a spiral stream. Figure 1 also shows the plot for [Cl–Cl-∗-Cl]^−^ (**16**) of the wide range of data.

Data from the fully optimized structures are analyzed using the polar coordinate (*R*, *θ*) representation [1,2], which corresponds to the static nature of the interactions. Each interaction plot, which contains data from both the perturbed and fully optimized structures, includes a specific curve that provides important information about the interaction.

This plot is expressed by (*θ*_p_, *κ*_p_), where *θ*_p_ corresponds to the tangent line of the plot, and *κ*_p_ is the curvature [1,2]. *θ* and *θ*_p_ are measured from the *y*-axis and *y*-direction, respectively. The parameters are illustrated in Figure 1a, exemplified by Br-∗-Br (**29**). *R*, *θ*, *θ*_p_ and *κ*_p_ are defined by Equations (4)–(7), respectively, and are given by the energy unit.

The concept of the dynamic nature of interactions has been proposed based on (*θ*_p_, *κ*_p_). The (*R*, *θ*) and (*θ*_p_, *κ*_p_) parameters are called the QTAIM-DFA parameters here [1,2,3]. The signs of the first derivatives of *H*_b_(***r***_c_) − *V*_b_(***r***_c_)/2 and *H*_b_(***r***_c_) (derived from (*H*_b_(***r***_c_) − *V*_b_(***r***_c_)/2)/d*r* and *H*_b_(***r***_c_)/d*r*, respectively, where *r* is the interaction distances in question) are used in the prediction of the natures of the interactions, in addition to those of *H*_b_(***r***_c_) − *V*_b_(***r***_c_)/2 and *H*_b_(***r***_c_). As a result, QTAIM-DFA incorporates the classification of the interactions with the QTAIM approach.
*R* = (*x*^2^ + *y*^2^)^1/2^(4)
*θ* = 90° − tan^−1^ (*y*/*x*)(5)
*θ*_p_ = 90° − tan^−1^ (d*y*/d*x*)(6)
*κ*_p_ = |d^2^*y*/d*x*^2^|/[1 + (d*y*/d*x*)^2^]^3/2^(7)
where (*x*, *y*) = (*H*_b_(***r***_c_) − *V*_b_(***r***_c_)/2, *H*_b_(***r***_c_)).

The reliability of the dynamic nature is controlled by the quality of the perturbed structures. We proposed a method to generate perturbed structures of excellent quality for QTAIM-DFA [7]. The method is called CIV, which employs the coordinates derived from the compliance constants *C_ii_* for the internal vibrations [8,9]. The dynamic nature of interactions based on the perturbed structures with CIV is described as the “intrinsic dynamic nature of interactions,” as the coordinates are invariant with the choice of the coordinate system. QTAIM-DFA is applied to the standard interactions, employing the perturbed structures generated with CIV, and rough criteria that distinguish the interaction in question from others are obtained. QTAIM-DFA and the criteria are explained in the Appendix A using Figure A1, Figure A2 and Figure A3, Figure A1 and Figure A2, and Table A1. The basic concept of the QTAIM approach is also explained [1,2,3].

Cremer has suggested that if bonding is investigated with the Laplacian of *ρ*_b_(***r***_c_) via Equation (8) (both sides of Equation (3) were multiplied by 2), the situation is not clearly covered when 2*G*_b_(***r***_c_) > |*V*_b_(***r***_c_)| > *G*_b_(***r***_c_). Therefore, he has suggested that in such cases, it seems to be more appropriate to choose *H*_b_(***r***_c_) as the indicator of the binding interactions [10]. Our proposed QTAIM-DFA method is a powerful tool that covers his proposal well and can also clearly separate vdW, HB, CT-MC, X_3_^–^ and CT-TBP by using perturbation structures. QTAIM-DFA has excellent potential in evaluating, classifying, characterizing, and understanding weak to strong interactions according to a unified form.
2*G*_b_(***r***_c_) + *V*_b_(***r***_c_) = (*ћ*^2^/4*m*)∇^2^*ρ*_b_(***r***_c_)(8)

The *θ*_p_ value of an interaction is usually larger than the corresponding *θ* value, and it is rare that the *θ*_p_ value is less than the *θ* value [11]. As shown in Figure 1, each plot for **1**–**29** seems almost parallel, albeit partially, to the plot of the wide range of **16**, although the plot for Me_2_Se^+^-∗-Cl (**26**) seems somewhat different from others. All interactions in Figure 1 are expected to show behavior that is very similar to the behavior of **16**, except for **26**. The behavior was examined based on the Δ*θ*_p_ (= *θ*_p_ − *θ*) values. The Δ*θ*_p_ values were positive for all interactions, except for C-∗-H in H_3_C-∗-H (**35**), for which the value was calculated to be Δ*θ*_p_ = –0.2° (< 0°). The prediction of the negative Δ*θ*_p_ value to **35** seems curious at first glance. It is not easy to image the result based on its plot, perhaps due to the very small magnitude. One could find a very slight difference in the plot for **35** from the plots for **33** and **36**. One could find the similarity to the difference of the plot for **26** from the plot for **25**, for example. In the case of **26**, the Δ*θ*_p_ value was calculated to be a positive value of 0.6°; Δ*θ*_p_ = 0.6° (> 0°). The magnitude is also very small. The interactions of Me_2_Se^+^-∗-Cl (**26**) and H_3_C-∗-H (**35**) should be on the borderline to give the positive and negative Δ*θ*_p_ values, judging from the magnitudes of Δ*θ*_p_. The Δ*θ*_p_ values for **1**–**36** are also collected in Appendix A. (See Figure 2 for the plot of Δ*θ*_p_ versus *θ* for **1**–**36**, except for **34**.)

Why is *θ*_p_ usually larger than *θ*? What is the origin of the positive and negative Δ*θ*_p_ values for interaction? What are the mechanisms that cause the positive and negative Δ*θ*_p_ values? Some expectations have been raised that it may be possible to elucidate the behavior of the interactions based on the Δ*θ*_p_ values. Then, we began to search for interactions that have negative Δ*θ*_p_ values by examining and reexamining the interactions, including those we investigated thus far.

Indeed, the standard interactions in **1**–**36** consist of the atoms of the first to fourth periods, but some interactions are expected to have negative Δ*θ*_p_ values if the interactions contain the atoms of the fifth period. Investigations are started by examining the nature of the interactions containing the atoms of the fifth period and F (**37**–**61**), which we call new standard interactions. Some interactions are found to have negative Δ*θ*_p_ values. We propose the concept of the normal behavior of interactions for Δ*θ*_p_ > 0 and the inverse behavior when Δ*θ*_p_ < 0. The origin of the negative Δ*θ*_p_ values is clarified by analyzing the QTAIM-DFA plots for the interactions with Δ*θ*_p_ < 0 over a wide range of interaction distances. The mechanisms to show the negative Δ*θ*_p_ values are also elucidated by analyzing the behavior of QTAIM functions over a wide range of interaction distances. Based on the analysis of various interactions. The behavior of the interactions with the negative Δ*θ*_p_ values is discussed after the application to the wider range of interactions. The inverse behavior of interactions is demonstrated to arise when *H*_b_(***r***_c_) − *V*_b_(***r***_c_)/2, and the corresponding kinetic energy densities at BCPs, *G*_b_(***r***_c_) do not show normal behavior. The results of the investigations are discussed.

## 2. Methodological Details of the Calculations

Gaussian 03 and 09 programs were employed for the calculations [12,13]. The aug-cc-pVTZ and/or 6-311+G(3df,3pd) basis sets were applied to the atoms of the group’s first–fourth elements in the calculations. They are called basis set system A (BSS-A) and basis set system B (BSS-B), respectively. The Sapporo-TZP basis sets with diffusion functions of the 1s1p type (abbreviated as S-TZPsp) were implemented from the Sapporo Basis Set Factory, which was called BSS-C [14,15]. The S-TZPsp basis sets were applied to the atoms of the fifth period in addition to BSS-A and BSS-B, which were called BSS-A’ and BSS-B’, respectively. The Møller–Plesset second-order energy correlation (MP2) level [16] was applied to the calculations. The S-TZPsp basis sets were used for I, Br and N with the 6-311+G(d,p) basis sets for F, O, S, C and H, which was called BSS-C’. The reaction processes for CF_3_SO_2_NXMe (X = Br and I) were calculated with MP2/BSS-C’. The optimized structures were confirmed by all real frequencies. The results were used to obtain the compliance constants (*C_ii_*) and the coordinates corresponding to *C_ii_* (**C***_i_*) [7,8,9]. The optimizations were not corrected with the BSSE method.

Equation (9) explains the method used to generate the perturbed structures with CIV [7]. The *k*th perturbed structure in question (**S***_kw_*) is generated by the addition of the coordinates corresponding to *C_ii_* (**C***_i_*) to the standard orientation of a fully optimized structure (**S**_o_) in the matrix representation. The coefficient *g_kw_* in Equation (9) controls the structural difference between **S***_kw_* and **S**_o_ [7]. The *g_kw_* is determined to satisfy Equation (10) for *r*, where *r* and *r*_o_ are the interaction distances in question in the perturbed and fully optimized structures, respectively. The **C***_i_* values of five digits are used to calculate **S***_kw_*.
**S***_kw_* = **S**_o_ + *g_kw_*•**C***_i_*(9)
*r* = *r*_o_ + *wa*_o_ (*w* = (0), ±0.05 and ±0.1; *a*_o_ = 0.52918 Å)(10)
*y* = *c*_o_ + *c*_1_*x* + *c*_2_*x*^2^ + *c*_3_*x*^3^ (*R*_c_^2^: square of correlation coefficient)(11)

The perturbed structures are also obtained by the partial optimization method (POM), where the interaction distances in question are fixed according to Equation (10), containing a wide range of *w* [1,2]. The reliability of the perturbed structures with POM is substantially the same as the reliability with CIV. The IRC (intrinsic reaction coordinates) method was also applied to generate the perturbed structures, starting from the transition states, TS.

QTAIM functions were calculated with the same method as the optimizations, unless otherwise noted. The calculated values were analyzed with the AIM2000 [17] and AIMAll [18] programs. *H*_b_(***r***_c_) is plotted versus *H*_b_(***r***_c_) − *V*_b_(***r***_c_)/2 for the data of five points of *w* = 0, ±0.05 and ±0.1 in Equation (10) in QTAIM-DFA. Each plot is analyzed using a regression curve of the cubic function, shown in Equation (11), where (*x*, *y*) = (*H*_b_(***r***_c_) − *V*_b_(***r***_c_)/2, *H*_b_(***r***_c_)) (*R*_c_^2^ > 0.99999 is typical) [3].

## 3. Results and Discussion

### 3.1. Basic Trend in the Δθ_p_ Values

The *H*_b_(***r***_c_) − *V*_b_(***r***_c_)/2 and *H*_b_(***r***_c_) values of the QTAIM functions and the QTAIM-DFA parameters of (*R*, *θ*) and (*θ*_p_, *κ*_p_) [1,2] for **1**–**36** (Figure 1) calculated under MP2/BSS-A are collected in Appendix A, together with the Δ*θ*_p_ and *C_ii_* values and the predicted nature. The Δ*θ*_p_ values are positive for all standard interactions of **1**–**36**, except for C-∗-H of H_3_C-∗-H (**35**), for which (*θ*, Δ*θ*_p_) = (202.8°, –0.2°). It is reasonable to assume that an interaction shows normal behavior if Δ*θ*_p_ > 0, whereas it does the inverse behavior when Δ*θ*_p_ < 0, which we propose in this work. Namely, all standard interactions in **1**–**36** behave normally, except for **35**, which shows inverse behavior, although the magnitude of Δ*θ*_p_ is very small. To elucidate the nature of the interactions based on the Δ*θ*_p_ values, investigations are started by examining the basic trend in the standard interactions of **1**–**36**, employing the Δ*θ*_p_ values.

Figure 2 shows the plot of Δ*θ*_p_ versus *θ* for **1**–**36** (Figure 1). The plot is analyzed using a regression curve of the eighth-order function after the addition of some fictional points, such as (*θ*, Δ*θ*_p_) = (45°, 0°) and (206.6°, 0°), and omitting the data for **35** and Me_2_Se^+^-∗-Cl (**26**). Fortunately, a smooth regression curve, described by *f*(Δ*θ*_p_), was obtained, and the curve passes very close to the points of (45°, 0°) and (206.6°, 0°). The regression curve is shown by a black solid line in Figure 2. The maximum point of the regression curve, shown by the solid line, was (*θ*, Δ*θ*_p_) = (109.7°, 48.9°). The regression curve was revised to a new curve by amplifying the maximum value of (109.7°, 48.9°) to (109.7°, 50.0°). The new curve is described by *f_r_*(Δ*θ*_p_) and shown by the dotted line. The treatment helps the discussion to be simpler, because the maximum value of 48.9° should be tentative and change depending on the employed species. Two more curves of *f_r_*(Δ*θ*_p_)/2 and −*f_r_*(Δ*θ*_p_)/2 are added in Figure 2, which are shown by the blue and red dotted lines, respectively, for a better explanation of the behavior.

The data for the normal behavior of the interactions with Δ*θ*_p_ > 0 appear on the upper side of the *x*-axis in Figure 2. Interactions show inverse behavior if the (*θ*, Δ*θ*_p_) points appear downside of the *x*-axis. The typical data for the normal behavior of interactions should appear around the black dotted line of *f_r_*(Δ*θ*_p_), whereas data for the inverse behavior of interactions appear around the red dotted line of −*f_r_*(Δ*θ*_p_)/2. An interaction is called to show weak normal behavior if the point appears between *f_r_*(Δ*θ*_p_)/2 and the *x*-axis. The interaction in Ne-∗-HF (**2**) seems to show borderline to weak normal behavior (Figure 2). While the interactions in **26** are borderline between weak normal and inverse behavior, that in **35** show inverse behavior, close to weak normal behavior.

Very similar results were obtained in the plot of Δ*θ*_p_ versus *θ* for **1**–**36** when calculated with BSS-B (see Appendix A for the data calculated with BSS-B). The Δ*θ*_p_ values are similarly plotted versus *θ* for the wide range of data of [Cl–Cl-∗-Cl]^−^ (**16**), calculated under aug-cc-pVTZ, where the perturbed structures of **16** are generated with POM. The plot is shown in Appendix A, together with the data for **1**–**36**. Figure A1 is essentially the same as Figure 2 (see Figure 1 for the QTAIM-DFA plot of **16**, shown by the dotted line). A detailed comparison of the two figures revealed that the plots are very close to each other when the Cl-∗-Cl distances in question are longer than the optimized values, whereas the Δ*θ*_p_ values for **16** are evaluated to be somewhat larger than those expected based on the data for **1**–**36** when the Cl-∗-Cl distances in question become shorter than the optimized values of 2.300 Å. When the interaction distance becomes shorter than the optimized value, the energy curve sharply tightens. The calculation conditions for POM under the interaction distances are widely shorter than the optimized values, where the shortened distance are 0.5 Å for 20 data points. The very severe conditions would be responsible for the results.

The close similarity between the two figures can be explained as follows: Starting from interatomic distances that are long enough, chemical bonds or interactions form as the distances shorten. The processes for the interactions are similar to each other, if the interactions of the normal behavior are compared. The processes can be understood based on the *H*_b_(***r***_c_) and *H*_b_(***r***_c_) − *V*_b_(***r***_c_)/2 values or the plot of *H*_b_(***r***_c_) versus *H*_b_(***r***_c_) − *V*_b_(***r***_c_)/2. In these cases, the (*θ*, *θ*_p_) values are very close to each other, if they are calculated at the points of substantially the same positions on the plots. Namely, starting from interaction distances that are far enough, stable interactions form as if they go along the similar plot of *H*_b_(***r***_c_) versus *H*_b_(***r***_c_) − *V*_b_(***r***_c_)/2. The drive on the plot arrives at a point of the minimum energy of the interaction, where the (*θ*, *θ*_p_) values are given for the interaction. Different (*θ*, *θ*_p_) values are obtained because the minimum energy is mainly determined depending on the interacting atoms. There must also be some differences in the energy curves, depending on the characteristics of the interactions, which result in the somewhat different curves of the plots. This should be the reason that the data points of various interactions appear close to the curve for the plot of (*θ*, Δ*θ*_p_) of **16**.

The next investigation is to search for the interactions with negative Δ*θ*_p_ values after the establishment of the basic trend in the normal behavior of the standard interactions.

### 3.2. Behavior of Various Interactions Based on θ and Δθ_p_

Table 1 collects the *H*_b_(***r*_c_**) − *V*_b_(***r*_c_**)/2 and *H*_b_(***r***_c_) values for **37**–**61**, similarly calculated under MP2/BSS-A’. Figure 3 shows the plots of *H*_b_(***r***_c_) versus *H*_b_(***r***_c_) − *V*_b_(***r*_c_**)/2 for the data of **37**–**61** shown in Table 1, where the perturbed structures are generated with CIV. Contrary to the case of Figure 1, some plots for the interactions in Figure 3 show different streams from the main (averaged) stream, which must be a reflection of the different behaviors of the interactions. The (*R*, *θ*) and (*θ*_p_, *κ*_p_) values were similarly calculated for **37**–**61** under MP2/BSS-A’. Table 1 collects the values, together with the Δ*θ*_p_ and *C_ii_* values and the predicted natures. The Δ*θ*_p_ values were plotted versus *θ* for the data of **37**–**61** shown in Table 1. Figure 4 shows the plots.

As shown in Figure 4 (and Table 1), the data of the (*θ*, Δ*θ*_p_) values drop very near the *f_r_*(Δ*θ*_p_) curve for most of **37**–**61**, which shows the normal behavior of the interactions, although some points appear much further below the *f_r_*(Δ*θ*_p_) curve. The data of the (*θ*, Δ*θ*_p_) values appear between the *f_r_*(Δ*θ*_p_)/2 curve and the *x*-axis for Me_2_Te-∗-F_2_ (**41**; (*θ*, Δ*θ*_p_) = (131.0°, 5.0°)), Me_2_BrTe-∗-Br (**51**; (173.1°, 3.5°)), Me_2_Se^+^-∗-I (**54**; (185.0°, 2.9°)) and Me_2_Te^+^-∗-I (**58**; (187.3°, 2.3°)). The interactions show weak normal behavior. The (*θ*, Δ*θ*_p_) points appear between the *x*-axis and the −*f_r_*(Δ*θ*_p_)/2 curve for F-∗-IF^−^ (**46**; (*θ*, Δ*θ*_p_) = (134.5°, −5.5°)), Me_2_FTe-∗-F (**49**; (122.6°, −3.0°)), Me_2_ClTe-∗-Cl (**50**; (166.3°, −6.6°)), Me_2_S^+^-∗-I (**53**; (181.7°, −4.0°)) and Me_2_Te^+^-∗-F (**55**; (126.0°, −2.4°)), which show the inverse behavior of the interactions. In the case of Me_2_Te^+^-∗-Cl (**56**; *θ*, Δ*θ*_p_) = (168.0°, −23.9°)), Me_2_Te^+^-∗-Br (**57**; (179.5°, −14.0°)), H_3_C-∗-F (**60**; (177.6°, −27.0°)) and H_3_C-∗-I (**61**; (187.2°, −9.5°)), the data appear below the −*f_r_*(Δ*θ*_p_)/2 curve. Therefore, the interactions are recognized to show inverse behavior stronger than that supposed from the behavior of the standard interactions in **1**–**36**.

What are the differences in the nature of the interactions among the three groups? The (*θ*, Δ*θ*_p_) values are examined in more detail for the interactions shown in Table 1. The typical (*θ*, Δ*θ*_p_) values for Me_2_Te-∗-X_2_ (**41**–**44**), Me_2_(X)Te-∗-X (**49**–**52**), Me_2_Te^+^-∗-X (**55**–**58**) and H_3_C-∗-X (**30**, **31**, **60**, and **61**) (X = F, Cl, Br, and I) are shown in Equations (12)–(15), respectively. The interactions in the equations are arranged in ascending order of Δ*θ*_p_.
(*θ*, Δ*θ*_p_) = (152.4°, 30.2°) for Me_2_Te-∗-I_2_ (**44**) > (164.7°, 20.6°) for Me_2_Te-∗-Br_2_ (**43**) >
(169.1°, 14.7°) for Me_2_Te-∗-Cl_2_ (**42**) > (131.0°, 5.0°) for Me_2_Te-∗-F_2_ (**41**)(12)
(176.7°, 11.1°) for Me_2_(I)Te-∗-I (**52**) > (173.1°, 3.5°) for Me_2_(Br)Te-∗-Br (**51**) >
(122.6, −3.0°) for Me_2_(F)Te-∗-F (**49**) > (166.3°, −6.6°) for Me_2_(Cl)Te-∗-Cl (**50**)(13)
(187.3°, 2.3°) for Me_2_Te^+^-∗-I (**58**) > (126.0°, −2.4°) for Me_2_Te^+^-∗-F (**55**) >
(179.5°, −14.0°) for Me_2_Te^+^-∗-Br (**57**) > (168.0°, −23.9°) for Me_2_Te^+^-∗-Cl (**56**)(14)
(193.8°, 5.2°) for H_3_C-∗-Cl (**30**) > (191.8°, 4.5°) for H_3_C-∗-Br (**31**) >
(187.2°, −9.5°) for H_3_C-∗-I (**61**) > (177.6°, −27.0°) for H_3_C-∗-F (**60**)(15)

Figure 5 shows the plot of Δ*θ*_p_ versus X (X = F, Cl, Br, and I) for **41**–**44**, **49**–**52**, **55**–**58** and **30**, **31**, **60**, and **61**. The Δ*θ*_p_ values for the interactions in **41**–**44**, **49**–**52** and **55**–**58** are in the order of Me_2_Te-∗-X_2_ > Me_2_(X)Te-∗-X > Me_2_Te^+^-∗-X, as a whole, if those of the same X are compared. The strength of Te–X is in the order of Me_2_Te-∗-X_2_ < Me_2_(X)Te-∗-X < Me_2_Te^+^-∗-X; therefore, the strength of Te–X in the species operates to decrease the Δ*θ*_p_ values for the species.

The Δ*θ*_p_ values for the interactions in **41**–**44** become more positive, as X goes from F to Cl, then Br, and then to I, where the character (or atomic number) of X becomes closer to Te in **41**–**44**. The Δ*θ*_p_ values in **49**–**52** and **55**–**58** show the trends, similar to that in **41**–**44**, for X = Cl, Br and I. The strength of Te–X operates to decrease the Δ*θ*_p_ values also for the species. However, the Δ*θ*_p_ values in **49** and **55** become more negative, when X goes F to Cl, contrary to the case in **41**. The differences in the strength of Te–F in **41**, **49** and **55** would result in smaller differences in the plot. In the case of H_3_C-∗-X, the Δ*θ*_p_ values become more negative in the order of X = Cl ≥ Br > I >> F. The behavior of the interactions should be controlled by the differences in the characters (or atomic numbers) of X. The character of C in H_3_C-∗-X should be closer to those of Cl and Br, relative to the case of I and F. The calculated results are well explained based on Equations (5) and (6).

What are the behavior of the E-∗-E’ bonds in the neutral, monoanionic, monocationic, and dicationic forms of [MeE-∗-E’Me]* (E, E’ = S, Se, and Te; * = null, −, +, and 2+)? The species are optimized, first. The optimized structures of the neutral form have *C*_2_ symmetry for E = E’ and close to *C*_2_ symmetry for E ≠ E’. Here, the structures are called the *C*_2_ type. The structures of the *C*_2_ and *trans* types are optimized for the monoanionic form. The structures of the *C*_2_ type are only discussed here because the Δ*θ*_p_ values are larger than 20° for both forms (Table 2). The structures of the *trans* type are optimized for the mono- and dicationic forms. Table 2 collects the (*θ*, *θ*_p_, Δ*θ*_p_) values of the species calculated under MP2/BSS-C. Numbers for the species are shown in Table 1. The calculated results, other than those in Table 2, are collected in Appendix A.

The behavior of the interactions is examined next, based on the (*θ*, Δ*θ*_p_) values. Positive Δ*θ*_p_ values of 28.2–32.3° are predicted for the monoanionic forms of [MeE-∗-E’Me]^−^ (E, E’ = S, Se and Te) shown in Table 2. Positive Δ*θ*_p_ values (2.6–6.8°) are predicted for S-∗-S, S-∗-Se, Se-∗-Se and Te-∗-Te of the neutral, mono- and dicationic forms of [MeE-∗-E’Me]* (E, E’ = S, Se, and Te; * = null, +, and 2+). However, negative Δ*θ*_p_ values are predicted for the neutral forms of MeS-∗-TeMe (Δ*θ*_p_ = −14.2°) and MeSe-∗-TeMe (−3.7°); the monocationic forms of [MeS-∗-TeMe]^+^ (c−15.5°) and [MeSe-∗-TeMe]^+^ (−3.9°); and the dicationic forms of [MeS-∗-TeMe]^2+^ (−11.3°) and [MeSe-∗-TeMe]^2+^ (−3.2°).
Se-∗-Se (6.6 ≤ Δ*θ*_p_ ≤ 6.8°) > S-∗-S (5.5°–5.8°) > S-∗-Se (2.6–6.5°) > Te-∗-Te (2.7–3.2°) >
Se-∗-Te (−3.7 − −3.2°) > S-∗-Te (−15.5 − −11.3°)(16)

Equation (16) shows the common order for Δ*θ*_p_ of E-∗-E’ in the neutral, mono- and dicationic forms of [MeE-∗-E’Me]* (E, E’ = S, Se, and Te), where the equation is arranged in the ascending order of Δ*θ*_p_. The order can also be understood under the guidance of the conclusions derived from Equations (5) and (6). That is, the Δ*θ*_p_ values are more negative as the difference in the atomic numbers of A and B in A-∗-B becomes larger. However, it is not so clear that the Δ*θ*_p_ values for the same A-∗-B become more negative if the interaction in question becomes stronger.

Figure 6 shows the plot of Δ*θ*_p_ versus *θ* for the E-∗-E’ interactions in the neutral, monoanionic, monocationic and dicationic forms of [MeEE’Me]* (E, E’ = S, Se and Te; * = null, −, + and 2+). (See Table 2 for the values). The normal, weak normal and inverse behavior of interactions is clearly illustrated in the figure. A very similar trend is observed in the neutral, anionic and dicationic forms of [HE-∗-E’H]* (E, E’ = S, Se and Te; * = null, − and 2+). The calculated results are collected in Appendix A, and the values are plotted in Figure 6. Some (data) points of (Δ*θ*_p_, *θ*) appear below the –*f*_r_(Δ*θ*_p_)/2 curve for S-∗-Te in 3E_Me_^0^, 3E_H_^0^, 3E_Me_^+^ and 3E_Me_^2+^, and 3E_H_^2+^ with Se-∗-Te in 5E_Me_^0^ and 5E_H_^0^. The results show that some S-∗-Te and Se-∗-Te interactions show stronger inverse behavior than that supposed from the behavior for the standard interactions in **1**–**36**.

The behavior of E-∗-E’ in [HE-∗-E’H]* (E, E’ = S, Se and Te; * = null, − and 2+) is very similar to the behavior in [MeE-∗-E’Me]* (E, E’ = S, Se and Te; * = null, −, + and 2+); therefore, the behavior of E-∗-E’ in [HE-∗-E’H]* (E, E’ = S, Se and Te; * = null, −, + and 2+) is not discussed in detail here. Nevertheless, the behavior of [HS-∗-TeH]^2+^ is discussed later.

The (*θ*, *θ*_p_, Δ*θ*_p_) values for E-∗-C and E’-∗-C, together with E-∗-H and E’-∗-H, in [REE’R]* (R = Me and H: E, E’ = S, Se, and Te; * = null, −, +, and 2+) are collected in Appendix A. Figure 7 shows the plot of Δ*θ*_p_ versus *θ*, for the data in Appendix A. A lot of data are well visualized.

As shown in the table and the figure, the Δ*θ*_p_ values are negative for Te-∗-C in MeETe-∗-Me (E = S, Se and Te: −24.5° ≤ Δ*θ*_p_ ≤ −23.5°), [MeETe-∗-Me]^–^ (E = S, Se and Te: −23.5° ≤ Δ*θ*_p_ ≤ −22.5°), [MeETe-∗-Me]^+^ (E = S, Se and Te: −23.1° ≤ Δ*θ*_p_ ≤ −22.3°), and [MeETe-∗-Me]^2+^ (E = S, Se and Te: −19.4° ≤ Δ*θ*_p_ ≤ −17.3°), whereas the values are all positive for S-v-C of [Me-∗-SE’Me]* (E’ = S, Se and Te; * = null, –, + and 2+: 3.9° ≤ Δ*θ*_p_ ≤ 5.4°). In the case of Se-∗-C, the Δ*θ*_p_ values are negative for MeESe-∗-Me (E = S, Se and Te: −1.4° ≤ Δ*θ*_p_ ≤ −0.7°) and [MeESe-∗-Me]^−^ (E = S, Se and Te: −2.8° ≤ Δ*θ*_p_ ≤ −2.5°), whereas the values are positive for [MeESe-∗-Me]^+^ (E = S, Se and Te: 1.4° ≤ Δ*θ*_p_ ≤ 2.2°) and [MeESe-∗-Me]^2+^ (E = S, Se and Te: 3.7° ≤ Δ*θ*_p_ ≤ 4.1°). The Δ*θ*_p_ values of E-∗-C and E’-∗-C are plotted in [Me-∗-EE’-∗-Me]* (E, E’ = S, Se and Te; * = null, −, + and 2+), containing the values from the symmetric [Me-∗-EE-∗-Me]* (E = S, Se and Te; * = null, −, + and 2+). In the case of E-∗-H and E’-∗-H, the Δ*θ*_p_ values are negative for all S-∗-H, Se-∗-H, and Te-∗-H shown in the table. The Δ*θ*_p_ values are −4.5° ≤ Δ*θ*_p_ ≤ −2.5° for HES-∗-H and [HES-∗-H]^−^, −13.9° ≤ Δ*θ*_p_ ≤ −12.2° for HESe-∗-H and [HESe-∗-H]^−^ and −20.9° ≤ Δ*θ*_p_ ≤ −17.1° for HETe-∗-H and [HETe-∗-H]^−^, where E = S, Se and Te. Most of the (data) points of (Δ*θ*_p_, *θ*) for E-∗-H (E = S, Se, and Te) and Te-∗-H appear below the −*f*_r_(Δ*θ*_p_) curve, tentatively drawn in the plot. The interactions show much severer the inverse behavior is than that supposed from the behavior for the standard interactions of **1**–**36**.

After recognizing the interactions with Δ*θ*_p_ < 0, the next extension is to clarify the origin and mechanisms that cause the negative Δ*θ*_p_ values.

### 3.3. Requirements for the Positive to Negative Values of Δθ_p_ for the Interactions in Question

The Δ*θ*_p_ value for an interaction is Δ*θ*_p_ > 0, Δ*θ*_p_ = 0 or Δ*θ*_p_ < 0, depending on the dynamic nature of the interaction. The dynamic nature was considered based on the characteristics of the plot around the BCP. Figure 8 illustrates the requirements for the QTAIM-DFA plot of *H*_b_(***r***_c_) versus *H*_b_(***r***_c_) − *V*_b_(***r***_c_)/2, where the arrows on the plot lines indicate the direction in which the interaction distance becomes shorter. As shown by the black plot in Figure 8, the Δ*θ*_p_ value for an interaction is positive when its plot line crosses the line for *θ* in the direction of clockwise rotation, viewed from the origin. The Δ*θ*_p_ value is negative when its plot line crosses the line for *θ* in the direction of counterclockwise rotation. The Δ*θ*_p_ value is zero when it is parallel to the line for *θ*, as shown by the blue plot.

### 3.4. Processes to Arise the Negative Δθ_p_ Values around the Optimized Structures

The processes to give the negative Δ*θ*_p_ values are examined for some interactions from the sufficiently distant states to around the optimized structures. *H*_b_(***r***_c_) values are plotted versus *H*_b_(***r***_c_) − *V*_b_(***r***_c_)/2 for the reaction processes over wide ranges of the interaction distances in question. It is more suitable if the singlet state is retained throughout the process, which relieves us from the trouble of considering the effect of the change on the multiplicity in the process. The structural change is also expected to be limited to the minimum extent if the singlet state is retained throughout the processes. The analysis is much simpler under these conditions, together with the discussion. As a result, the mechanisms to give the negative Δ*θ*_p_ values are more clearly understood as the reactions proceed if the singlet state is retained throughout the reaction processes. Such processes seem rather rare and satisfy the above conditions.

The negative Δ*θ*_p_ value of −4.2° is calculated for [F–I-∗-F]^−^ (**46**), as shown in Table 1, where the reaction process is shown by Equation (17). The reaction shown by Equation (18) has been reported to proceed in the singlet state under metal-free conditions [19,20]. A negative value of −22.8° was also calculated for Tf–N-∗-IMe (Tf = SO_2_CF_3_) (Table 3). The reaction processes for [F–I-∗-F]^–^ (**46**) in Equation (17) and Tf–N-∗-IMe in Equation (18) could be good candidates to achieve this purpose. Therefore, the reaction processes to give the negative Δ*θ*_p_ values are examined, exemplified by the reactions shown in Equations (17) and (18).
F–I + F^−^ → [F–I-∗-F]^−^(17)
Tf–N-∗-BrMe + MeI → TS [MeBr-∗-N(Tf)-∗-IMe] → MeBr + Tf–N-∗-IMe(18)
Tf–N-∗-BrMe + MeBr → TS [MeBr-∗-N(Tf)-∗-BrMe] → MeBr + Tf–N-∗-BrMe(19)

Figure 9 shows the QTAIM-DFA plot of *H*_b_(***r***_c_) versus *H*_b_(***r***_c_) − *V*_b_(***r***_c_)/2 for [F–I-∗-F]^−^ for a wide range of the interaction distances in question. Fortunately, the perturbed structures were successfully (and easily) obtained for a wide range of interaction distances under MP2/BSS-C. Figure 9 contains the plot for [Cl–Cl-∗-Cl]^−^, similarly calculated, for comparison (see also Table 1). The plot for [Cl–Cl-∗-Cl]^−^ shows a smooth and monotonic curve, starting from a point close to the origin. The plot for [Cl–Cl-∗-Cl]^−^ forms a spiral curve overall. As a result, the curve of the plot for [Cl–Cl-∗-Cl]^−^ satisfies the requirements for *θ*_p_ > *θ* (Δ*θ*_p_ > 0) throughout the reaction process, as illustrated in Figure 9. This is the reason that [Cl–Cl-∗-Cl]^−^ typically shows the normal behavior of interactions.

However, the plot for [F–I-∗-F]^−^ seems different from the plot for [Cl–Cl-∗-Cl]^−^ in some points. The plot does not show the spiral figure as a whole. The plot of [F–I-∗-F]^−^ seems similar to the plot of [Cl–Cl-∗-Cl]^−^ when *H*_b_(***r***_c_) − *V*_b_(***r***_c_)/2 < 0.01 au, as shown in the magnified Figure 9b. However, the plots for the two interactions become much different if *H*_b_(***r***_c_) − *V*_b_(***r***_c_)/2 > 0.02 au. The *H*_b_(***r***_c_) value seems to decrease almost linearly as the *H*_b_(***r***_c_) − *V*_b_(***r***_c_)/2 value becomes larger, especially for the range of *H*_b_(***r***_c_) − *V*_b_(***r***_c_)/2 > 0.02 au. The plot for [F–I-∗-F]^−^ seems to show a downwardly convex character for the range, which is very different from the plot for [Cl–Cl-∗-Cl]^−^. The characteristic figure in the plot for [F–I-∗-F]^−^ prevents us from showing the spiral stream. The curve for the plot of [F–I-∗-F]^−^ would satisfy the requirements for *θ*_p_ > *θ* (Δ*θ*_p_ > 0) in the range of *H*_b_(***r***_c_) − *V*_b_(***r***_c_)/2 < 0.02 au, as illustrated in Figure 9. The curve gradually changes to show the downwardly convex character to satisfy the requirements for *θ*_p_ = *θ* (Δ*θ*_p_ = 0). The optimized structure, shown by *w* = 0.0 in Figure 9, appears in the range satisfying the requirements for *θ*_p_ < *θ* (Δ*θ*_p_ < 0), illustrated in Figure 8. Indeed, the plot in Figure 9 shows the outline to give the negative Δ*θ*_p_ value (–5.5°) for [F–I-∗-F]^−^ (**46**), but it is better if the requirement for Δ*θ*_p_ < 0 is illustrated more clearly in the plot for a process. Next, this process was examined and exemplified by the ligand exchange reaction at N via TS [MeBr-∗-N(Tf)-∗-IMe].

Table 3 collects the (*θ*, *θ*_p_, Δ*θ*_p_) values for the interactions of Tf–N-∗-BrMe, Tf–N-∗-IMe and TS [MeBr-∗-N(Tf)-∗-IMe], which have been calculated at the BCPs under MP2/BSS-C’. Table 3 contains the values for TS [MeBr-∗-N(Tf)-∗-BrMe], similarly calculated, for reference. The perturbed structures were calculated in the processes shown in Equations (17) and (18) with the IRC method, starting from TS [MeBr-∗-N(Tf)-∗-IMe] and TS [MeBr-∗-N(Tf)-∗-BrMe]. The perturbed structures of the Br-∗-N and N-∗-I interaction distances of wide ranges could be successfully calculated with the IRC method under MP2/BSS-C’.

The reaction processes shown in Equation (19) were examined before investigations on those shown in Equation (18). *H*_b_(***r***_c_) is plotted versus *H*_b_(***r***_c_) − *V*_b_(***r***_c_)/2 for both Br-∗-N bonds in the ligand exchange reaction via TS [MeBr-∗-N(Tf)-∗-BrMe]. The plot is shown in Appendix A, consisting of two plots, very close to each other, because TS [MeBr-∗-N(Tf)-∗-BrMe] has no symmetry due to the slightly different geometry around the Br-∗-N-∗-Br moiety. Therefore, the nature of Br-∗-N on both sides of the TS [MeBr-∗-N(Tf)-∗-BrMe] is slightly different. The Δ*θ*_p_ values are calculated to be 50.3° and 49.1° for TS [MeBr-∗-N(Tf)-∗-BrMe], respectively.

The plots for the two N-∗-Br interactions via TS [MeBr-∗-N(Tf)-∗-BrMe] show a smooth, monotonic and spiral curve, very similar to the curve for [Cl–Cl-∗-Cl]^–^ (Figure 1 and Figure 9). Consequently, both Br-∗-N are recognized to show the typical normal behavior of interactions, which leads to positive Δ*θ*_p_ values for the final products. The Δ*θ*_p_ values of the final products of MeBr-∗-N–Tf and the isomer are 7.4° and 5.8°, respectively, depending on slight differences in the geometry. The similarity in the reaction processes between MeBr-∗-N–Tf and [Cl-∗-Cl-∗-Cl]^−^ is of significant interest. The processes proceed in both directions and give the final products from TS [MeBr-∗-N(Tf)-∗-BrMe] of the highest energy for the former but from the global minimum of the lowest energy for the latter.

Figure 10 shows the QTAIM-DFA plots for the process shown by Equation (18). Figure 10 is briefly explained to avoid misunderstanding. It is not a plot for the energy profile, but a QTAIM-DFA plot for both Br-∗-N and N-∗-I, so two plots appear corresponding to the two bonds, while a single transition state contributes to the reaction. (This is also the case for the plots in Appendix A.) The forward and reverse processes for the reaction shown by Equation (18) are clearly specified in Figure 10.

In spite of the forward and reverse processes for the reaction, the plot for Br-∗-N is drawn by the black dots in Figure 10, which corresponds to the reaction process of Equation (18). The process forms MeI + Tf–N-∗-BrMe, starting from Tf–N-∗-IMe + MeBr via TS [MeBr-∗-N(Tf)-∗-IMe]. The Δ*θ*_p_ value of Br-∗-N changes from 7.9° for Tf–N-∗-BrMe to 42.6° for TS [MeBr-∗-N(Tf)-∗-IMe], then to 0.0°, with no interaction state, if the reaction proceeds in the direction shown in Equation (18).

The plot for Br-∗-N in Figure 10 also shows a smooth, monotonic and spiral curve, which is very similar to the curve for [Cl–Cl-∗-Cl]^−^ (Figure 1 and Figure 9) and the curves for Br-∗-N illustrated in Appendix A (see Equation (19)). Consequently, the Br-∗-N bond in the ligand exchange reactions via TS [MeBr-∗-N(Tf)-∗-IMe] shows normal behavior similar to the case of TS [MeBr-∗-N(Tf)-∗-BrMe] and [Cl–Cl-∗-Cl]^−^. However, the plot for I-∗-N in Figure 10 is drawn by the red dots. The process forms MeBr + Tf–N-∗-IMe, starting from Tf–N-∗-BrMe + MeI via TS [MeBr-∗-N(Tf)-∗-IMe]. The Δ*θ*_p_ value of I-∗-N goes from 0.0° for MeI to 48.4° for TS [MeBr-∗-N(Tf)-∗-IMe], then to −22.8° for Tf–N-∗-IMe if the reaction proceeds in the direction shown in Equation (18). The plot for I-∗-N is very different from the plot for N-∗-Br, as shown in Figure 10. The difference between the plots clarifies the mechanism to give the negative Δ*θ*_p_ value for Tf–N-∗-IMe.

The plot for N-∗-I in red shows a distorted Z shape. This is of great interest because the plot for N-∗-I seems to largely overlap the plot for N-∗-Br in the range of *H*_b_(***r***_c_) > −0.02 au, if the plot for N-∗-I is moved to the right. The plot for N-∗-I curves to the left in the range of −0.05 < *H*_b_(***r***_c_) < −0.006 au, then crosses the plot for N-∗-Br at *H*_b_(***r***_c_) around −0.07 au. The plot for N-∗-I shows a downward-sloping linear shape in the range of *H*_b_(***r***_c_) < –0.08 au. The mechanism to cause the negative Δ*θ*_p_ value for Tf–N-∗-IMe, based on the characteristic figure in the plot of N-∗-I shown in Figure 10, can be explained as follows: The Δ*θ*_p_ value for N-∗-I in the reaction, shown in Equation (18), must be positive if *H*_b_(***r***_c_) > −0.06 au for N-∗-I, starting from Tf–N-∗-BrMe + MeI (no interaction between N and I). The positive Δ*θ*_p_ value decreases for the plot curving to the left in the range of −0.05 < *H*_b_(***r***_c_) < −0.01 au, then approaches zero, and finally gives the negative Δ*θ*_p_ value of −12.9° for Tf–N-∗-IMe in the range of *H*_b_(***r***_c_) < −0.09 au.

In the plot for N-∗-I in Figure 10, *H*_b_(***r***_c_) decreases monotonically, whereas *H*_b_(***r***_c_) – *V*_b_(***r***_c_)/2 shows abnormal behavior when *H*_b_(***r***_c_) < 0. Therefore, it is strongly suggested that the strange behavior of *H*_b_(***r***_c_) – *V*_b_(***r***_c_)/2 should be responsible for the negative Δ*θ*_p_ value.

The process for N-∗-I to give the negative Δ*θ*_p_ value is well clarified by the plot for N-∗-I in Figure 10. However, it would be difficult to visually recognize the value of Δ*θ*_p_, positive or negative, from the plot. We searched for a typical example that can be recognized as a clear negative Δ*θ*_p_ value visually from the plot. The (*θ*, *θ*_p_, Δ*θ*_p_) values for [HS-∗-TeH]^2+^ are given in Appendix A, of which Δ*θ*_p_ is a negative value of −9.7°. The process for [HS-∗-TeH]^2+^ was calculated with POM by elongating the S-∗-Te distance, starting from the structure around the optimized one. The calculations were performed in the range where the rational structures were optimized at the singlet state.

Figure 11 draws the reaction process for [HS-∗-TeH]^2+^, although the interaction distance is limited to the distance around the optimized structure, where the perturbed structures are rationally optimized at the singlet state. The plots for N-∗-I and N-∗-Br in the reaction processes shown in Equation (18) are added to Figure 11 for reference. The partial plot for [HS-∗-TeH]^2+^, shown in Figure 11, seems to correspond to the plot for N-∗-I in the range of *H*_b_(***r***_c_) < −0.005 au. The lines of Δ*θ*_p_ = 0° for the plots of N-∗-I and [HS-∗-TeH]^2+^ are drawn by the red and blue dotted lines, respectively. The tangent directions from the origin to the curves correspond to the lines of Δ*θ*_p_ = 0° (see also Figure 8).

As discussed above, the plot for N-∗-Br satisfies the requirement for Δ*θ*_p_ > 0° in the whole range of the reaction process, while the plot for N-∗-I should satisfy the requirement for Δ*θ*_p_ < 0° around the optimized structure. However, it seems visually unclear from the plot, as mentioned above. In the case of the plot for [HS-∗-TeH]^2+^, there exists an area that clearly satisfies the requirements for Δ*θ*_p_ < 0°, after the blue dotted line for Δ*θ*_p_ = 0°. The area with Δ*θ*_p_ < 0° is the area around the optimized structure in this case. The crossing point on the plot with the tangent line corresponds to Δ*θ*_p_ = 0°; therefore, the perturbed structure of which the S-∗-Te distance is shorter than the value at Δ*θ*_p_ = 0° should show positive Δ*θ*_p_ values, namely the area of Δ*θ*_p_ ≥ 0°.

After clarification of the origin for the positive and negative Δ*θ*_p_ values through the plots, the next extension is to elucidate the mechanisms based on the behavior of *G*_b_(***r***_c_) and *V*_b_(***r***_c_).

### 3.5. Mechanisms for the Origin of the Negative Δθ_p_ Values

The reaction processes can be directly expressed using the interaction distances. However, the *ρ*_b_(***r***_c_) values can also be used for the analyses, where the values decrease exponentially as the interaction distances increase. The distances are also expressed by *w* in Equation (10). The relation between *ρ*_b_(***r***_c_) and *w* is confirmed by the plot of *ρ*_b_(***r***_c_) versus *w* for an interaction. Such a plot is shown in Appendix A, exemplified by [Cl–Cl-∗-Cl]^−^ and [F–I-∗-F]^−^. As a result, *ρ*_b_(***r***_c_) would be the better parameter to describe the characteristics of interactions around the optimizes structures, in more detail, relative to the case of *r* and *w* for the longer distances. The characteristic behavior of *H*_b_(***r***_c_) versus *H*_b_(***r***_c_) − *V*_b_(***r***_c_)/2 can be attributed to the behavior of *G*_b_(***r***_c_) and *V*_b_(***r***_c_), because *H*_b_(***r***_c_) and *H*_b_(***r***_c_) − *V*_b_(***r***_c_)/2 are expressed by *G*_b_(***r***_c_) and *V*_b_(***r***_c_) (see Equations (1) and (2)). The reaction processes are investigated employing the plots of *H*_b_(***r***_c_) − *V*_b_(***r***_c_)/2, *H*_b_(***r***_c_), *G*_b_(***r***_c_) and *V*_b_(***r***_c_) versus *ρ*_b_(***r***_c_) and the related plots.

Figure 12a,b shows the plots of *H*_b_(***r***_c_) − *V*_b_(***r***_c_)/2, *H*_b_(***r***_c_), *G*_b_(***r***_c_) and *V*_b_(***r***_c_) versus *ρ*_b_(***r***_c_) for the reaction processes of [Cl–Cl-∗-Cl]^−^ and [F–I-∗-F]^−^, where the perturbed structures are generated with POM over the wide range of *w*. The plots of *G*_b_(***r***_c_) versus *ρ*_b_(***r***_c_) and *V*_b_(***r***_c_) versus *ρ*_b_(***r***_c_) for [Cl–Cl-∗-Cl]^−^ go upside and downside, respectively, very smoothly and monotonically, starting from a point close to the origin. The plot of the former goes upside almost linearly, while that of the latter goes downside, showing a slight convex upward shape. The magnitude of the former seems close to but less than two times the magnitude of the latter for the range of *ρ*_b_(***r***_c_) < 0.16 (*e*/*a*_0_^3^). The plots of *H*_b_(***r*_c_**) − *V*_b_(***r*_c_**)/2 versus *ρ*_b_(***r*_c_**) and *H*_b_(*r*_c_) versus *ρ*_b_(***r***_c_) are also smooth and gentle for [Cl–Cl-∗-Cl]^−^. The *H*_b_(***r***_c_) − *V*_b_(***r*_c_**)/2 value is positive and then negative after *ρ*_b_(***r***_c_) ≥ 0.16 (*e*/*a*_0_^3^), while the positive *H*_b_(***r***_c_) becomes negative when *ρ*_b_(***r***_c_) reaches approximately 0.026 (*e*/*a*_0_^3^), leading to the normal behavior of the [Cl–Cl-∗-Cl]^−^ interactions (Δ*θ*_p_ > 0). The plot of *H*_b_(***r***_c_) versus *H*_b_(***r***_c_) − *V*_b_(***r***_c_)/2 for [Cl–Cl-∗-Cl]^−^ shown in Figure 9 is well explained through the plots of *H*_b_(***r***_c_) − *V*_b_(***r***_c_)/2 versus *ρ*_b_(***r***_c_) and *H*_b_(***r***_c_) versus *ρ*_b_(***r***_c_) shown in Figure 12a.

In the case of [F–I-∗-F]^−^, the plots of *G*_b_(***r***_c_) versus *ρ*_b_(***r***_c_) and *V*_b_(***r***_c_) versus *ρ*_b_(***r***_c_) go upside and downside, respectively, smoothly and monotonically, starting from a point close to the origin. However, the plots show convex upward and downward shapes, respectively, and the magnitude of the former seems 0.8 times larger than the magnitude of the latter. As a result, the plot of *H*_b_(***r***_c_) − *V*_b_(***r***_c_)/2 versus *ρ*_b_(***r***_c_) is mainly controlled by the plot of *G*_b_(***r***_c_) versus *ρ*_b_(***r***_c_), whereas the plot of *H*_b_(***r***_c_) versus *ρ*_b_(***r***_c_) is almost controlled by the plot of *V*_b_(***r***_c_) versus *ρ*_b_(***r***_c_) (cf; Equations (1) and (2)). *H*_b_(***r***_c_) − *V*_b_(***r***_c_)/2 and *H*_b_(***r***_c_) are positive and negative in the calculated whole range for [F–I-∗-F]^−^, except for *H*_b_(***r***_c_) of the substantially large range of *r* where *ρ*_b_(***r***_c_) is substantially very small. Therefore, the plot of *H*_b_(***r***_c_) versus *H*_b_(***r***_c_) − *V*_b_(***r***_c_)/2 is close to the linear relationship of the *y* = −*a*x (*a* > 0) type, which essentially explains the plot for [F–I-∗-F]^−^, as shown in Figure 9. The character of *G*_b_(***r***_c_) versus *ρ*_b_(***r***_c_) remains in the character of *H*_b_(***r***_c_) − *V*_b_(***r***_c_)/2 versus *ρ*_b_(***r***_c_) for [F–I-∗-F]^−^. The negative Δ*θ*_p_ value of [F–I-∗-F]^−^ must arise from the characteristic plot of *H*_b_(***r***_c_) versus *H*_b_(***r***_c_) − *V*_b_(***r***_c_)/2, which is close to the linear relationship of the *y* = −a*x* (*a* > 0) type, which originates from the characteristic plot of *H*_b_(***r***_c_) versus *G*_b_(***r***_c_).

Figure 12c,d show the plots of d(*H*_b_(***r***_c_) − *V*_b_(***r***_c_)/2)/d*ρ*_b_(***r***_c_), d*H*_b_(***r***_c_)/d*ρ*_b_(***r***_c_), d*G*_b_(***r***_c_)/d*ρ*_b_(***r***_c_) and d*V*_b_(***r***_c_)/d*ρ*_b_(***r***_c_) versus *ρ*_b_(***r***_c_) for the reaction process of [Cl–C-∗-Cl]^−^ and [F–I-∗-F]^−^. All plots seem almost linearly flat, except for the plots of d(*H*_b_(***r***_c_) − *V*_b_(***r***_c_)/2)/d*ρ*_b_(***r***_c_), d*G*_b_(***r***_c_)/d*ρ*_b_(***r***_c_) and d*V*_b_(***r***_c_)/d*ρ*_b_(***r***_c_) for [F–I-∗-F]^−^. The results for F–I-∗-F]^−^ expose the behavior in the plot of d(*H*_b_(***r***_c_) − *V*_b_(***r***_c_)/2)/d*ρ*_b_(***r***_c_) versus *ρ*_b_(***r***_c_). The behavior of the plot of *H*_b_(***r***_c_) − *V*_b_(***r***_c_)/2 versus *ρ*_b_(*r*_c_) is shown to be the main factor leading to the negative Δ*θ*_p_ value for F–I-∗-F]^−^, as expected.

Figure 13a,b illustrate the plots of *H*_b_(***r***_c_) − *V*_b_(***r***_c_)/2, *H*_b_(***r***_c_) *G*_b_(***r***_c_) and *V*_b_(***r***_c_) versus *ρ*_b_(***r***_c_) for N-∗-Br and N-∗-I in the reaction processes from TS [MeBr-∗-N(Tf)-∗-IMe], where the perturbed structures are generated with the IRC method (see Equation (18)).

The plots of *H*_b_(***r***_c_) − *V*_b_(***r***_c_)/2, *H*_b_(***r***_c_), *G*_b_(***r***_c_), and *V*_b_(***r***_c_) versus *ρ*_b_(***r***_c_) for N-∗-Br are very similar to the corresponding plots for [Cl–C-∗-Cl]^−^, although the magnitudes of the plots are different. Consequently, the plot of *H*_b_(***r***_c_) versus *H*_b_(***r***_c_) − *V*_b_(***r***_c_)/2 for N-∗-Br should be very similar to the plot for [Cl–C-∗-Cl]^−^. The plot for N-∗-Br is well explained based on the analyzed results discussed above.

However, the plots of *G*_b_(***r***_c_) versus *ρ*_b_(***r***_c_) and *V*_b_(***r***_c_) versus *ρ*_b_(***r***_c_) for N-∗-I go upside and downside, respectively, starting from a point close to the origin. The plot of *G*_b_(***r***_c_) versus *ρ*_b_(***r***_c_) for N-∗-I gives a distorted Z-shaped curve, while the plot of *V*_b_(***r***_c_) versus *ρ*_b_(***r***_c_) shows a slightly distorted Z-shaped character. The magnitude of the plot of *G*_b_(***r***_c_) versus *ρ*_b_(***r***_c_) seems to be 0.7 times the magnitude of *V*_b_(***r***_c_) versus *ρ*_b_(***r***_c_) for N-∗-I. Therefore, the plot of *H*_b_(***r***_c_) − *V*_b_(***r***_c_)/2 versus *ρ*_b_(***r***_c_) is controlled by the plot of *G*_b_(***r***_c_) versus *ρ*_b_(***r***_c_), whereas the plot of *H*_b_(***r***_c_) versus *ρ*_b_(***r***_c_) is controlled by the plot of *V*_b_(***r***_c_) versus *ρ*_b_(***r***_c_). *H*_b_(***r***_c_) − *V*_b_(***r***_c_)/2 and *H*_b_(***r***_c_) are positive and negative in the whole range calculated for N-∗-I, except for *H*_b_(***r***_c_) of the substantially very small range *ρ*_b_(***r***_c_). Therefore, the plot of *H*_b_(***r***_c_) versus *H*_b_(***r***_c_) − *V*_b_(***r***_c_)/2 appears in the regular CS region, except for the region of *H*_b_(***r***_c_) > 0, very close to the origin. The character of *G*_b_(***r***_c_) versus *ρ*_b_(***r***_c_) remains in the plot of *H*_b_(***r***_c_) − *V*_b_(***r***_c_)/2 versus *ρ*_b_(***r***_c_) for N-∗-I, again. The positive value of *H*_b_(***r***_c_) becomes negative at approximately 0.028 (*e*/*a*_0_^3^) for *ρ*_b_(***r***_c_), and then the value becomes more negative as *ρ*_b_(***r***_c_) becomes larger.

In the case of *H*_b_(***r***_c_) − *V*_b_(***r***_c_)/2, the value is positive in the whole region calculated for N-∗-I (Figure 13a). The plot of *H*_b_(***r***_c_) − *V*_b_(***r***_c_)/2 versus *ρ*_b_(***r***_c_) for N-∗-I gives a rather clear distorted Z-shaped curve. The plot goes upside after the start of the plot at around the origin, then it goes downside and then goes upside. Consequently, the Z-shaped figure of *H*_b_(***r***_c_) versus *H*_b_(***r***_c_) − *V*_b_(***r***_c_)/2 for N-∗-I shown in Figure 10 is well analyzed based on the characteristic behavior of the plot of *H*_b_(***r***_c_) − *V*_b_(***r***_c_)/2 versus *ρ*_b_(***r*_c_**), together with the plot of *H*_b_(***r***_c_) versus *ρ*_b_(***r***_c_). The abnormal behavior of *H*_b_(***r***_c_) − *V*_b_(***r***_c_)/2, derived from the abnormal behavior of *G*_b_(***r***_c_), must be responsible for the negative Δ*θ*_p_ value of N-∗-I in CF_3_SO_2_N-∗-IMe.

Figure 13c,d show the plots of d(*H*_b_(***r***_c_) − *V*_b_(***r***_c_)/2)/d*ρ*_b_(***r***_c_), d*H*_b_(***r***_c_)/d*ρ*_b_(***r***_c_), d*G*_b_(***r***_c_)/d*ρ*_b_(***r***_c_) and d*V*_b_(***r***_c_)/d*ρ*_b_(***r***_c_) versus *ρ*_b_(***r***_c_) for the process for N-∗-Br and N-∗-I in TS [MeI–N(Tf)-∗-BrMe]. The characteristic behaviors in the plots of *H*_b_(***r***_c_) − *V*_b_(***r***_c_)/2, *H*_b_(***r***_c_) *G*_b_(***r***_c_) and *V*_b_(***r***_c_) versus *ρ*_b_(***r***_c_) is be emphasized by the plots of d(*H*_b_(***r***_c_) − *V*_b_(***r***_c_)/2)/d*ρ*_b_(***r***_c_), d*H*_b_(***r***_c_)/d*ρ*_b_(***r***_c_), d*G*_b_(***r***_c_)/d*ρ*_b_(***r***_c_) and d*V*_b_(***r***_c_)/d*ρ*_b_(***r***_c_) versus *ρ*_b_(***r***_c_). The plots of d*G*_b_(***r***_c_)/d*ρ*_b_(***r***_c_) and d*V*_b_(***r***_c_)/d*ρ*_b_(***r***_c_) versus *ρ*_b_(***r***_c_) for N-∗-Br show concave and convex figures, respectively, while the plots for N-∗-I show apparent wave figures. The plots of d*G*_b_(***r***_c_)/d*ρ*_b_(***r***_c_) versus *ρ*_b_(***r***_c_) and d*V*_b_(***r***_c_)/d*ρ*_b_(***r***_c_) versus *ρ*_b_(***r***_c_) behave oppositely for the both cases of N-∗-Br and N-∗-I. The magnitudes of the changes in the plots of d*G*_b_(***r***_c_)/d*ρ*_b_(***r***_c_) and d*V*_b_(***r***_c_)/d*ρ*_b_(***r***_c_) versus *ρ*_b_(***r***_c_) are larger for N-∗-I than for N-∗-Br.

The plot of d*H*_b_(***r***_c_)/d*ρ*_b_(***r***_c_) versus *ρ*_b_(***r***_c_) for N-∗-I is very close to the plot for N-∗-Br, which decreases very smoothly and monotonically with increasing magnitude as *ρ*_b_(***r***_c_) becomes larger due to the expected relation of d*H*_b_(***r***_c_)/d*ρ*_b_(***r***_c_) = d*G*_b_(***r***_c_)/d*ρ*_b_(***r***_c_) + d*V*_b_(***r***_c_)/d*ρ*_b_(***r***_c_). However, the plot of d(*H*_b_(***r***_c_) − *V*_b_(***r***_c_)/2)/d*ρ*_b_(***r***_c_) versus *ρ*_b_(***r***_c_) for N-∗-I shows a concave then convex (wave) figure. The character in the plot of d*G*_b_(***r***_c_)/d*ρ*_b_(***r***_c_) versus *ρ*_b_(***r***_c_) must remain in the plot of d(*H*_b_(***r***_c_) − *V*_b_(***r***_c_)/2)/d*ρ*_b_(***r***_c_) versus *ρ*_b_(***r***_c_), mainly due to the expected relation of d(*H*_b_(***r***_c_) − *V*_b_(***r***_c_)/2)/d*ρ*_b_(***r***_c_) = d*G*_b_(***r***_c_)/d*ρ*_b_(***r***_c_) + d*V*_b_(***r***_c_)/2d*ρ*_b_(***r***_c_). The results support the following statements: The inverse behavior of the N-∗-I interaction must originate based on the convex then concave curve in the plot of *H*_b_(***r***_c_) − *V*_b_(***r***_c_)/2 versus *ρ*_b_(***r***_c_). The loose Z-shaped plot of *H*_b_(***r***_c_) versus *H*_b_(***r***_c_) − *V*_b_(***r***_c_)/2 for N-∗-I is observed, starting from a point close to the origin. The N-∗-Br and N-∗-I interactions are demonstrated through the above discussion, which show typical normal and inverse behavior, respectively.

Applications of the proposed methodology to wider range of bonds and interactions, such as a bird’s-eye view of the periodic table, are in progress. The results will be discussed elsewhere, together with the results, related to ours, reported so far by others.

## 4. Conclusions

The QTAIM-DFA parameters of (*R*, *θ*) and (*θ*_p_, *κ*_p_) are obtained by analyzing the QTAIM-DFA plots of *H*_b_(***r***_c_) versus *H*_b_(***r***_c_) − *V*_b_(***r***_c_)/2. The *θ*_p_ value is usually larger than *θ*, Δ*θ*_p_ (= *θ*_p_ − *θ*) > 0, for an interaction, as confirmed by the standard interactions, but it is sometimes negative. The prediction of the nature of interactions is confused when Δ*θ*_p_ < 0 because the criteria to predict the nature are formulated assuming positive Δ*θ*_p_ values. The negative Δ*θ*_p_ value for an interaction must be a sign of its special nature. We searched for such interactions that show negative Δ*θ*_p_ values. Negative Δ*θ*_p_ values are typically detected for [F–I-∗-F]^−^, [RE-∗-TeR]* (R = Me and H; E = S and Se; * = null, +, and/or 2+) and CF_3_SO_2_N-∗-IMe, although the Δ*θ*_p_ values are typically positive for most interactions. Asterisks are used to emphasize the existence of BCPs on the BPs. We propose the concept of the normal and inverse behavior of interactions for Δ*θ*_p_ > 0 and Δ*θ*_p_ < 0, respectively.

The Δ*θ*_p_ value for an interaction is positive when its plot line crosses the line for *θ* in the direction of clockwise rotation, viewed from the origin. The Δ*θ*_p_ value is negative when its plot line crosses the line for *θ* in the direction of counterclockwise rotation. The Δ*θ*_p_ value is zero when it is parallel to the line for *θ*, as shown by the blue plot (see Figure 5). The characteristic behavior, normal or inverse, of interactions is elucidated by plotting *H*_b_(***r***_c_) versus *H*_b_(***r***_c_) − *V*_b_(***r***_c_)/2 for the reaction process containing [Cl–Cl-∗-Cl]^−^, TS [MeBr–N-∗-BrMe(SO_2_CF_3_)], [F–I-∗-F]^−^, HS-∗-TeH^2+^ and TS [MeBr-∗-N–IMe(SO_2_CF_3_)]. The Δ*θ*_p_ values are shown to be positive for Cl-∗-Cl and Br-∗-N but negative for I-∗-F, S-∗-Te and N–I. The abnormal character of *H*_b_(***r***_c_) − *V*_b_(***r***_c_)/2, derived from the abnormal character of *G*_b_(***r***_c_), must be responsible for the negative Δ*θ*_p_ values of the interactions. The large differences in the atomic numbers between the interacting atoms seem to greatly affect behavior. It is also very important to clarify the role of the differences in the atomic numbers on the behavior of the interactions.

## Data Availability

Not applicable.

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
