# Peer review of "Inverse Versus Normal Behavior of Interactions, Elucidated Based on the Dynamic Nature with QTAIM Dual-Functional Analysis"

_ijms, 2023, doi:10.3390/ijms24032798_

Round 1

Reviewer 1 Report

The manuscript by W. Nakanishi et al. is devoted to further development of QTAIM dual function analysis (QTAIM-DFA). This method allows to classify interactions using interrelation between two QTAIM parameters, the total energy density and the Laplacian of electron density. I think the manuscript should be published in the International Journal of Molecular sciences, since the developed method classifies interactions in an unambiguous manner. The manuscript needs only a minor revision, mainly technical. In my opinion the following issues should be addressed:

1) I think the two parameters examined in the study, namely the angle between the x-axis and the tangent and the curvature should be defined in the Introduction (definitions are given in Table 1 caption when the results are analysed).

2) The Authors may consider modifying the way they present their results, following the paper often needs moving back and forth from the main article to the Electronic Supplementary Information. I understand that the Authors present a huge amount of data but still I think it could be shown in a more reader-friendly manner.

Minor technical issues:

1) line 47 “shard shell” should be “shared-shell”.

Author Response

Thank you very much for your comments and advice. We revised the MS according to your comments and advice.

1) I think the two parameters examined in the study, namely the angle between the x-axis and the tangent and the curvature should be defined in the Introduction (definitions are given in Table 1 caption when the results are analyzed).

-->

Figure 1a (species 29), explanatory sentence, and Equations (3)–(6) were added.

L.79–L.81:

θ and θp are measured from the y-axis and y-direction, respectively. The parameters are illustrated in Figure 1a, exemplified by Br-*-Br (29). R, θ, θp and κp are defined by Equations (3)–(6), respectively and given in the energy unit.

L.90–L.94:

R = (x2 + y2)1/2                                             (3)

θ = 90° – tan–1 (y/x)                                    (4)

θp = 90° – tan–1 (dy/dx)                             (5)

kp = Ιd2y/dx2Ι/[1 + (dy/dx)2]3/2               (6)

where (x, y) = (Hb(rc) – Vb(rc)/2, Hb(rc))

2) The Authors may consider modifying the way they present their results, following the paper often needs moving back and forth from the main article to the Electronic Supplementary Information. I understand that the Authors present a huge amount of data but still I think it could be shown in a more reader-friendly manner.

-->

We have tried to make the contents understandable only by the text.

L.71–L.72: Chart 1. Species and compound numbers

L.79–L.81: q and qp are measured from the y-axis and y-direction, respectively. The parameters are illustrated in Figure 1a, exemplified by Br-*-Br (29). R, θ, θp and κp are defined by Equations (3)–(6), respectively and given in the energy unit.

L.90–L.94: Equations (3)–(6) were added.

In addition, the MS was revised and made more readable, which is indicated in the text by marking it in light blue. We have also made some innovations to the Figures.

3) We could not use the "Track Changes" function, so we marked the changes with light blue markers.

4) We checked that all references are relevant to the contents of the manuscript.

5) We used an English editing service to check our manuscript.

Sincerely yours,     

Satoko Hayashi

Reviewer 2 Report

This is an important contribution towards understanding bonding and interactions using a modified QTAIM technique. It may be accepted for publications after a minor revision.

1. The authors should compare their approach with that reported in CROATICA CHEMICA ACTA CCACAA 57 (6) 1259-1281 (1984)

2. Number of self-citations may be reduced.

3. They should analyze the Pauli repulsion.

Author Response

Reply to Referee #2:

Thank you very much for your comments and advice. We revised the MS according to your comments and advice.

  1. The authors should compare their approach with that reported in CROATICA CHEMICA ACTA CCACAA 57 (6) 1259-1281 (1984).

-->

Cremer's achievement is great. We had not been able to locate the Croatian paper. We were able to obtain it this time because of your advice. Cremer's proposal and ours are very similar. Sentences were added to the text as follows. 

(We are preparing a paper on the inverse behavior of interactions for all combinations of atoms in the periodic table. We will make a detailed comparison with Cremer's work.)

New ref. 7 was added.

Cremer, D.; Kraka E., Croatica Chemica Acta. CCACCA, 1984, 57, 1259–1281. [https://hrcak.srce.hr/file/286247]

L.107–L.116:

Cremer has suggested that if bonding is investigated with the Laplacian of ∇2ρb(rc) via Equation (2’’) (Both sides of Equation (2') were multiplied by 2), the situation is not clearly covered, when 2Gb(rc) > |Vb(rc)| > Gb(rc). Therefore, he has suggested that in such cases, it seems to be more appropriate to choose Hb(rc) as the indicator of the binding interactions [10]. Our proposed QTAIM-DFA method is a powerful tool that covers his proposal well and can also clearly separate vdW, HB, CT-MC, X3, and CT-TBP by using perturbation structures. QTAIM-DFA has excellent potential for evaluating, classifying, characterizing, and understanding weak to strong interactions according to a unified form.

2Gb(rc) + Vb(rc) = (ћ2/4m)∇2ρb(rc)                                        (2’’)

  1. Number of self-citations may be reduced.

-->

Original references 3 and 4 have been removed from the MS.

Reference numbering has been arranged.

  1. They should analyze the Pauli repulsion.

-->

We have added the following two sentences to the text.

L.727–L.729:

Applications of the proposed methodology to wider range of bonds and interactions, like a bird’s-eye view of the periodic table, are in progress. The results will be discussed elsewhere, together with the results, related to ours, reported so far by others.

We are investigating with Gaussian program, and Pauli repulsion requires the use of EDA software for GAMESS program.

We plan to investigate the possibility of obtaining and analyzing the Pauli repulsion in a future paper. This analysis was not performed at this time.

And as Cremer mentioned, we plan to study in detail not only the electron density, but also the inverse behavior factors in relation to the Gb(rc) and Vb(rc) energies.

  1. We have tried to make the contents understandable only by the text. In addition, the MS was revised and made more readable, which is indicated in the text by marking it in light blue. We have also made some innovations to the Figures.

  1. We could not use the "Track Changes" function, so we marked the changes with light blue markers.

  1. We checked that all references are relevant to the contents of the manuscript.

  1. We used an English editing service to check our manuscript.

Sincerely yours, 

Satoko Hayashi
